# Does the Pro-Environmental Behavior of Household PV Installation Contribute to the Shaping of Users’ Green Purchasing Behavior?—Evidence from China

**DOI:** 10.3390/bs13070612

**Published:** 2023-07-24

**Authors:** Shali Wang, Ruohan Zhang, Xiaodong Guo, Haijing Ma, Jiaxi Wu, Ying Wang, Shuangshuang Fan

**Affiliations:** 1School of Management, China University of Mining & Technology, Beijing 100083, China; teesn235@gues.edu.cn (S.W.); bqt2200502014@student.cumtb.edu.cn (R.Z.); bqt2000502011@student.cumtb.edu.cn (J.W.); 2School of Economics and Management, Guizhou University of Engineering Science, Bijie 551700, China; 3School of Accounting, Guizhou University of Finance and Economics, Guiyang 550025, China; guo517348019@163.com; 4Institute of Climate Change and Sustainable Development, Tsinghua University, Beijing 100084, China; mahaijing@tsinghua.edu.cn; 5School of Economics and Management, Wenzhou University of Technology, Wenzhou 325035, China

**Keywords:** household PV, green purchasing, promotion policies, generalized structural equation model, learning by doing

## Abstract

In order to achieve the “dual carbon goal”, the Chinese government is actively encouraging the adoption of household photovoltaic (PV) systems. While there has been considerable research on residents’ inclination to install PV, limited attention has been given to understanding how the installation and utilization of PV systems influence pro-environmental behaviors. Therefore, this paper aims to investigate the potential impact of pro-environmental behavior resulting from household PV installation on users’ green purchasing behavior. Based on the “learning by doing” theory, a survey was conducted with 1249 participants, and the generalized structural equation model was employed as our analytical approach. The findings of this research indicate that the adoption and utilization of household photovoltaic (PV) systems have a positive impact on green consumption. The test results demonstrate that the overall effect coefficient is 0.03, indicating that current PV promotion policies have an indirect impact on green consumption. Moreover, economic incentive policies have a more substantial influence than environmental publicity policies, with total indirect effect coefficients of 0.005 and 0.002, respectively. Based on the findings above, the following recommendations are proposed: (1) It is recommended to maintain stable economic incentives to promote the adoption of household PV systems. (2) Emphasizing the dissemination of knowledge and skills for promoting environmental protection should be prioritized. (3) Efforts should be made to align personal interests and societal interests with low-carbon policies.

## 1. Introduction

According to the “2020 Emissions Gap Report” by the United Nations Environment Programme, residential consumption accounts for two-thirds of global carbon emissions [1]. Research in China has further demonstrated that household consumption contributes over 50% of the country’s total carbon emissions, making it the second largest sector responsible for CO_2_ emissions [2]. Hence, it is argued by scholars that cultivating low-carbon, green, and environmentally friendly consumer behavior among the general public is crucial for China to attain its “Two carbon goals” of energy conservation and emissions reduction [3]. Therefore, this paper aims to explore the topic of green consumption among residents. Green consumption refers to consumer behavior that aims to reduce the negative environmental impact throughout the entire life cycle of products, including their purchase, use, and disposal, driven by environmental protection concerns [4]. The promotion of green consumption is essential in attaining sustainable development for humanity. It is important to note that human behavior is not inherent but rather influenced by the environment [5]. Similarly, the consumption of green products, as a manifestation of human behavior, is subject to various influencing factors. Therefore, this paper aims to delve into the underlying mechanisms that shape residents’ green consumption patterns. 

Currently, the Chinese government acknowledges the significance of transforming the lifestyles and energy consumption patterns of its residents in order to achieve the “Two Carbon Goals.” As a result, there is a strong push to promote residential photovoltaics (PV) and transition from traditional fossil fuels to renewable sources such as solar power [6]. To facilitate this, a comprehensive promotion plan has been developed [7]. Currently in China, residential PV installation primarily focuses on retrofitting rooftops and walls of residential buildings with PV systems [8]. It is important to highlight that the installation and utilization of residential PV not only entail transforming the living environment, but also entail reciprocal interactions with the surrounding residential environment. The concept of “material participation” suggests that residents’ behavior can be understood as an ongoing interaction with their immediate environment. As residents modify their surroundings, these changes, in turn, influence their thoughts and actions over time [9]. Similarly, the theory of “learning by doing” posits that the most effective means of learning is through hands-on practice [10]. The installation and utilization of household photovoltaics within the residential environment involve interactive processes. The daily management and maintenance of these devices can be considered as a green practice. Hence, we assert that the behavior of residents in installing and deploying household photovoltaics will progressively influence their attitudes and subsequent actions. Several studies have demonstrated the interrelation between pro-environmental behaviors A and B, with one behavior influencing the other [11,12]. In the current context, it is evident that the adoption of household photovoltaics in China is primarily motivated by external factors, such as government policies, which aim to accomplish the “dual carbon goals.” However, green consumption behavior is an intrinsically pro-environmental behavior that remains unaffected by external forces, making it an internal factor. Therefore, we can consider the installation and use of household photovoltaics by residents, driven by external factors, as A behavior, while green consumption behavior can be identified as B behavior. The presence of A behavior may potentially influence B behavior. 

According to statistics from the National Energy Administration of China, as of October 2021, the number of residential photovoltaic installations in China has surpassed 6 million households, impacting over tens of millions of individuals [13]. The cumulative installed capacity of residential photovoltaics has increased from 0.93 GW in 2016 to 20 GW in 2020 [7], demonstrating an average annual growth rate exceeding 40% [6]. To accelerate the uptake of residential photovoltaics, China released the “Notice on Submitting Pilot Program for Distributed Photovoltaic Development in Whole Counties (Cities, Districts)” on 24 June 2021 [7]. This initiative aims to facilitate the widespread adoption of residential photovoltaics within county-level cities by implementing appropriate policies. If the installation and utilization of residential photovoltaics can effectively shape residents’ green consumption behavior, it suggests that integrating government promotion policies into the installation and utilization of residential photovoltaics will not only achieve the desired behavior but also foster a ripple effect in promoting green consumption behavior. Hence, public policies have the potential to efficiently achieve a significant impact by capitalizing on the aforementioned effects. 

However, while many scholars have acknowledged the importance of governmental public management policies and residents’ environmental practices in achieving sustainable development, the majority of them view residents’ environmental practices as merely a collection of individual choices, with behaviors being seen as unrelated to one another [14,15]. This study investigates the potential causal relationships among individual environmental behaviors from a correlational perspective. It specifically examines the interrelationship between the installation of household PV systems and green consumption, expanding on previous studies that focused solely on single behaviors. By doing so, this research aims to contribute to the advancement of basic theories and research paradigms in the field of environmental behavior and management. This paper aims to investigate the link between policies promoting photovoltaic (PV) technology, PV installation rates, and residents’ green consumption behavior. Specifically, it seeks to address the following three research questions: (1)Is there a notable disparity in green consumption behavior between residents who install and utilize PV systems and those who do not?(2)If there is indeed a disparity as mentioned in question one, what factors contribute to this divergence?(3)In addition to encouraging PV installation, can China’s current PV promotion policy effectively stimulate green consumption behavior among residents through PV utilization?

Most of the current literature considers residents’ pro-environmental behaviors as separate systems [16]. While some studies focus on the factors that influence the adoption of household photovoltaic (PV) systems [17], and others examine the drivers of green consumption [4], there has been no previous research that combines these two aspects to investigate the effect of installing household PV on green consumption behavior. Considering this, the objective of this paper is to establish a connection between the installation and utilization of residential photovoltaic (PV) systems and residents’ green purchasing behavior. Additionally, it aims to investigate how the installation and utilization of household PV systems influence and shape residents’ green consumption habits. Secondly, this paper investigates the mechanisms through which the installation of PV systems influences green consumption by analyzing data obtained from 1249 questionnaires and employing a generalized structural equation model. Furthermore, this study examines the overall indirect impact of household PV promotion policies on fostering users’ green purchasing behavior. 

This paper is structured as follows: Section 2: Literature Review provides a comprehensive overview of relevant previous research, setting the foundation for the current study; Section 3: Research Design outlines the sources of data and the methods used for data analysis; Section 4: Research Results and Discussion presents the statistical analysis findings and offers a thorough discussion of the results; Section 5: Conclusion. In this section, the main research findings of this paper will be summarized and their significance elucidated, and an evaluation of the present study will be provided. 

## 2. Literature Review

A recent study has revealed an intriguing phenomenon known as “behavioral spillover” wherein residents’ participation in certain environmental behaviors affects their willingness or extent of involvement in other environmental actions [11,18]. Scholars have identified both positive and negative behavioral spillover effects. For example, research conducted through an experiment indicated that residents’ participation in water conservation activities led to a noteworthy 5.6% rise in weekly energy consumption [19]. A separate study revealed that engaging in household energy conservation measures led to a decrease in residents’ endorsement of government carbon tax policies, with an impact of up to 15% [20]. These research findings shed light on the precise cause-and-effect relationships between residents’ environmental behaviors. 

The concept of behavioral spillover provides researchers with a vital opportunity to observe and understand the intricate behavioral dynamics within individual environmental practices [21]. Traditional research in environmental behavior tends to emphasize the decision-making process of individuals regarding specific behaviors and views the outcomes of these behaviors as a result of individual psychological representations [22]. Conversely, behavioral spillover analysis suggests potential causal relationships between individual environmental behaviors and considers them as integral components of an interconnected and complex system. As a result, behavioral spillover analysis goes beyond conventional research paradigms, fostering theoretical innovation and advancing research frameworks in the field of environmental behavior and management [21]. This novel perspective enables a more thorough investigation into individual environmental practices, diverging from previous limited research perspectives. 

In the realm of residential photovoltaic (PV) adoption, current scholarly attention primarily centers on exploring residents’ inclination towards installing and utilizing PV systems [23,24]. Scholars analyze the factors that impact residents’ willingness to install from internal perspectives, including attitudes [14], perceptions [25], and motivations [26], as well as external factors such as government policies and living environment. There is a scarcity of literature that examines the nuanced impacts experienced by residents while using residential photovoltaics post-installation. Presently, no research has explored the social functions that arise from residents’ installation and utilization of residential photovoltaics. 

However, some scholars have directed their attention towards exploring the factors that drive green purchasing, such as consumer awareness [27], attitude [28], and economic power [29]. Alternatively, there are those who have approached the study of consumers’ motivation for green purchasing from a sociological standpoint, investigating the impact of face consciousness [30], mass media influence [31], and other related factors. All of the aforementioned studies have primarily focused on analyzing consumer green purchasing behavior within a limited scope, neglecting the integration of residents’ green purchasing behavior with other pro-environmental behaviors. 

The aforementioned studies present a substantial body of empirical evidence for examining residents’ pro-environmental behaviors. However, they fail to acknowledge the interconnectedness of residents’ behavioral decisions [32] and the influential role that environmental behaviors with longer practice periods have on shaping their other pro-environmental behaviors [33]. Therefore, this paper explores the relationship between residents’ adoption and utilization of household PV systems and their inclination towards green consumerism. In this study, we examine the relationship between the installation and use of household PV (photovoltaic systems) as a prior behavior, and consumers’ inclination towards green purchases as a subsequent behavior. Additionally, we incorporate government policies promoting household PV adoption into the model to analyze the indirect effect of these policies on residents’ green consumption. By analyzing the mechanism through which government policies influence residents’ green consumption via the promotion of PV installation, we aim to gain insights into the impact of such policies. Furthermore, building upon the prior analysis of the pertinent variables, residents’ policy perceptions will serve as a surrogate variable. The subsequent section will provide a comprehensive explanation of the particular research design. 

## 3. Research Design

### 3.1. Theoretical Analysis

According to learning theory, the process of learning involves attention, retention, repetition, and motivation reproduction [34]. Additionally, learning can encompass various forms such as social learning, school learning, and practical learning [35]. Practical learning, also known as the “learning by doing” model, refers to the accumulation of experience through the production of products and provision of services, leading to knowledge acquisition [10]. The acquisition of knowledge enhances an individual’s competence in addressing diverse challenges encountered in both professional and personal spheres. Bandura coined the term “self-efficacy” to describe this augmented sense of proficiency and self-assurance resulting from learning [34]. Since then, numerous learning theories have been applied in the field of environmental management. Chen et al. have introduced the term “green self-efficacy” to describe an individual’s competence and self-assurance in effectively addressing environmental concerns [36]. Since the installation and use of household PV is a long-term endeavor, it aligns with the “attention-retention-repetition-motivation reproduction” process elucidated by learning theory. Additionally, users engage in repeated practice throughout the entire process, which leads us to posit that the household PV installation process can indeed be viewed as a “learning by doing” experience. The installation of residential PV systems produces clean electricity and lowers carbon emissions, rendering it an “ecological practice”. Marres (2012) also points out that everyday consumer products in people’s homes possess social characteristics. By incorporating these tools into their daily routines, residents can actively engage with government policies or express their perspectives [9]. The environmentally friendly nature of residential photovoltaic (PV) systems inevitably attributes users as “environmental participants.” This association enhances users’ skills and self-assurance in addressing environmental concerns and boosts their sense of green self-efficacy. The practice of green consumption is considered as a pro-environmental action. We argue that the users’ belief in their own ability to engage in green behavior, also known as green self-efficacy, can effectively enhance their green consumption practices. In conclusion, our theoretical framework provides an explanation on how residents can enhance their green consumption levels by adopting and utilizing household photovoltaic systems.
(1)First, residents choose to install household PV under the stimulation of government policies;(2)The usage process after installation is seen as a “learning by doing” process and “material participation”, where users improve their green self-efficacy through “learning by doing” and “material participation”;(3)Finally, under the influence of green self-efficacy, the level of green consumption was improved.

In the subsequent sections of this paper, we will put forth specific pathway hypotheses grounded in the aforementioned procedure. 

### 3.2. Research Design and Research Methods

This paper aims to analyze using methods such as literature review, questionnaire survey, comparative study, generalized structural equation modeling (GSEM), and partial least squares structural equation modeling (PLS-SEM). 

The specific analysis process is as follows: Step 1: Induct from the literature to propose the research hypothesis and establish a conceptual model based on the hypothesis. Step 2: Using the conceptual model, induct from the literature to develop the research questionnaire. Step 3: Construct the questionnaire and collect data through a survey. Step 4: Establish a generalized structural equation model using the conceptual model and the collected data. Step 5: Conduct modeling analysis on the generalized structural equation model using PLS-SEM. 

The specific analysis procedure is as follows: the aforementioned analysis procedure is the structural equation analysis method recommended by Hair et al. [37], which has been widely used in existing literature and has high reliability and validity [38]. Therefore, this study adopts the aforementioned analysis procedure. 

The advantage of the literature review method in proposing research hypotheses and establishing conceptual models lies in the following aspects: Firstly, through literature review, researchers can make use of existing literature resources, avoiding duplicative research efforts, which effectively saves time and spatial costs in the research process. Additionally, literature review can help researchers quickly obtain a large amount of research information, enabling them to better understand and grasp the latest developments in the research field [39]. 

The questionnaire survey method has advantages such as high efficiency, convenience, wide applicability, and strong reliability. It is a commonly used research method that can be used to obtain a large amount of authentic and comparable research data. Therefore, we choose to collect data using the questionnaire survey method during the data collection process [40]. 

Comparative research method is a commonly used research approach, primarily utilized to compare the differences and similarities between two or more phenomena, events, groups, etc. [41]. The reason for adopting this method is that comparative research helps to identify causal relationships between different variables. By comparing the performance of variables under different conditions, it is possible to infer the causal connection between them, thus providing a better explanation of the reasons and outcomes of the research phenomenon [42]. In this study, we intend to conduct a comparative research between users who already have experience with residential photovoltaic installations and residents who do not have such experience, in order to analyze their differences and explore the causal relationships between variables. Therefore, we have chosen this method. 

The reason for choosing Generalized Structural Equation Modeling (GSEM) analysis method is that it allows for the consideration of relationships between multiple variables, including direct and indirect effects. It is suitable for analyzing data with multiple levels, groups, and variables, making it more appropriate for complex research questions [43]. GSEM provides an overall framework for assessing causal relationships between variables. By specifying the relationships between latent variables and observed variables, researchers can uncover the influence of latent factors on observed variables and explain the mechanisms behind these effects [44]. In this study, we plan to compare and explore causal relationships by contrasting users with experience in residential solar photovoltaic (PV) installation and residents without such experience, in order to examine the impact mechanisms of residential PV installation on residents’ green purchasing behavior. Therefore, we intend to construct a comparative variable: users with experience in residential PV installation (represented as 1) versus residents without such experience (represented as 0). This type of variable is known as a dummy variable and can only be estimated within the framework of a generalized structural equation model, thus justifying the use of GSEM for analysis. 

This study uses a generalized structural equation model for modeling and analysis. There are generally two methods for estimating the parameters of the structural equation model: the structural equation based on covariance estimation (CB-SEM) [45] and the structural equation based on partial least squares estimation (PLS-SEM) [46]. Researchers believe that these two methods complement each other. When the research model has been theoretically verified, the variables are reactive constructs, and the sample data conform to the normal distribution hypothesis, CB-SEM is more suitable for estimation [47]. When the research purpose is prediction or exploratory analysis, or the variables are formative constructs or mixed constructs, especially when the data do not necessarily conform to the normal distribution hypothesis, the PLS-SEM estimation procedure shall be used [48]. Hair et al. believe that the PLS-SEM estimation program is superior to CB-SEM [48]. The variables and models are not based on established theories, so this study is essentially exploratory research. Therefore, we choose PLS-SEM estimation program as the data analysis method of this study. 

In summary, the present study comprehensively considers factors such as research objectives, feasibility of implementation, reliability and validity of research results, extrapolation effects, etc., and ultimately selects the aforementioned combination of research methods. 

### 3.3. Formulation of Research Hypothesis

According to the theoretical elaboration above, we divide the hypothesis into four parts: part 1: the role of government policies on residents’ installation of household PV; part 2: the installation and use of household PV can bring users a sense of green efficacy; part 3: the sense of green efficacy can enhance residents’ green consumption; and part 4: the derivative hypothesis.

#### 3.3.1. The Impact of Government Policies on Residents’ Installation of Household PV

Social psychology suggests that the environment can influence individual decision-making [49], indicating that the environment in which individuals are situated can have an impact on their decision-making. Government policies are regarded as environmental factors that influence individual decision-making [50]. Government policies can affect individual decision-making, and this principle can also be applied to the field of pro-environmental behavior [51]. The above principle is commonly applied in research on pro-environmental behavior. This demonstrates that the influence of government policies on individual decision-making has been widely applied in the context of environmental conservation. Numerous studies have shown that government subsidies have a positive impact on the adoption of residential solar panels by residents [17]. This illustrates that government economic incentives (such as subsidies) can encourage residents to adopt residential solar panels. Government’s environmental promotional campaigns also facilitate the adoption of residential solar panels by residents [14,52]. Environmental promotion has a positive influence on individual decision-making. For the sake of measurement convenience, this study chooses to use policy perception as a substitute variable instead of policy measures. This allows for a more convenient measurement of the impact of policies on individual decision-making. This study uses economic incentive perception and environmental significance perception as substitute variables. Economic incentive perception refers to the incentives provided by the government for the installation of residential solar panels that residents perceive before the installation, including perception of economic subsidies and electricity cost benefits [7]. Environmental significance perception refers to residents’ perception of the environmental effects that may result from the installation and usage of residential solar panels after receiving environmental promotional information [53]. 

In summary, this study proposes the following hypotheses:

**H1a.** 
*The Perception of economic incentives can positively promote the installation of household PV by residents.*


**H1b.** 
*The Perception of environmental significance can positively promote the installation of household PV by residents.*


#### 3.3.2. The Impact of the Installation and Use of Household PV on Users’ Sense of Green Efficacy

According to the theory of experiential learning in human-centered design, practice is a form of learning [10] and learning can enhance self-efficacy [54]. Environmental practice is also a form of learning and can acquire environmental knowledge and skills, thereby enhancing green self-efficacy [55]. On the other hand, according to the theory of consumer value, consumers generate user experiences when using products or services, which include perceived value and perceived risk [7,56,57]. Particularly, when using technology-based durable goods such as automobiles, consumers experience richer perceived value and perceived risk [7,56]. Perceived value refers to the economic, functional, cognitive, and social value that residents perceive when using household photovoltaics. Perceived risk covers user perceptions of product quality, installation quality, potential safety risks, and policy risks. The user experience of using products and services can be viewed as a process of practice. As users can simultaneously perceive the value and risk of a product during practice, we can consider perceived value and perceived risk as substitute variables for user practice when using technology-based durable goods. Based on the theory of “learning through practice,” practice is considered as a learning process. Learning can enhance self-efficacy, and environmental practice can improve green self-efficacy. Therefore, we believe that the alternative variables of perceived value and perceived risk in practice can also enhance self-efficacy. Furthermore, as the installation and use of residential photovoltaics is a green practice, the corresponding perceived value and perceived risk contribute to green self-efficacy. To facilitate understanding, we illustrate with the example of consumers purchasing and using cars. After purchasing a car, consumers will gain a deeper understanding of the convenience brought by using the car (perceived value), while also facing risks such as traffic accidents and violations (perceived risk). In order to seek benefits and avoid harm, users need to improve their driving skills, learn traffic regulations, and identify road conditions, which is a continuous learning process through practice. Similarly, the installation and use of residential photovoltaics require users to enhance maintenance and management abilities to seek benefits and avoid harm. Therefore, we propose that perceived value and perceived risk can enhance green self-efficacy. Based on the above, this paper puts forward the following hypotheses:

**H1c.** 
*Residents can generate perceived value of PV equipment by installing and using household PV.*


**H1d.** 
*Residents can generate perceived risk of PV equipment by installing and using household PV.*


**H1e.** 
*Perceived value in user experience can enhance users’ green self-efficacy.*


**H1f.** 
*Perceived risk in user experience can enhance users’ green self-efficacy.*


#### 3.3.3. Green Efficacy Can Enhance Residents’ Green Consumption Level

Referring to previous literature and combining with the purpose of this study, green efficacy (GE) is defined as the residents’ confidence in human management of environmental problems and the evaluation of their ability to perform environmental behaviors, which specifically contains two dimensions: (1) confidence in human management of the environment and (2) evaluation of their ability to perform environmental behaviors [58,59]. Chen et al. argue that green efficacy enhances the level of individual environmental behaviors [36], and since green purchasing behavior belongs to an environmentally friendly behavior, the following hypothesis was derived:

**H1g.** 
*Green self-efficacy enhances the level of green purchasing of residents.*


#### 3.3.4. Derivativeness Hypothesis 

Based on the above hypotheses of Section 3.3.1, Section 3.3.2 and Section 3.3.3, the following hypotheses are proposed to be tested:(1)Total indirect effect hypothesis for green self-efficacy

**H2a.** 
*The installation and use of household PV can have a positive total indirect effect on enhancing users’ green self-efficacy. (Based on: H1c, H1d, H1e, H1f).*


**H2b.** 
*Perception of economic incentives can have a positive total indirect effect on enhancing users’ green self-efficacy. (Based on: H1a, H2a).*


**H2c.** 
*Perception of environmental significance can have a positive total indirect effect on enhancing users’ green self-efficacy. (Based on: H1b, H2a).*



(2)Total indirect effect hypothesis for green purchase


**H3a.** 
*The installation and use of household PV can have a positive total indirect effect on enhancing users’ green purchasing behavior. (Based on: H2a, H1g).*


**H3b.** 
*Perception of economic incentives can have a positive total indirect effect on enhancing users’ green purchasing behavior. (Based on: H2b, H1g).*


**H3c.** 
*Perception of environmental significance can have a positive total indirect effect on enhancing users’ green purchasing behavior. (Based on: H2c, H1g).*


### 3.4. Measurement of Variables

As shown in Appendix A. In this study, the data were collected using a questionnaire, and the following seven constructs were summarized based on the research purpose and path hypothesis presented above: perception of economic incentive (POEI), perception of environmental significance (POES), perceived value (PV), perceived risk (PR), green efficacy (GE), and green consumption (GC). Common method variance (CMV) may exist in the process of data collection using the questionnaire, which means that the overlap of variance between two variables is due to the use of similar measurement instruments rather than representing the true relationship between the underlying constructs. Since this study proposes to use a scale questionnaire to collect data in the quantitative analysis part, which is inevitably prone to common method variance error [60], we used a control variable method to measure and control the generation of common method variance, as suggested by Lindell et al. [61]. This method requires the design of a variable that is unrelated to the variables in the study model in terms of content, which is generally named: the marked variable (MV). This variable is incorporated into the model as a control variable during model construction and “controls” all endogenous variables in the model. The advantage of this method is that it allows the detection of common method variation and control of common method variation in one model, so we designed an MV using this method, which was originally used to measure the health awareness of the population and is not relevant to the model in this study in terms of content, so we selected it as the marked variable in this study. The seven constructs are listed in Table 1, along with the measurement items and literature sources. 

The questionnaire items were referred to existing literature. The Likert five-component scale was used to measure the values of the variables from 1 to 5: “strongly disagree”, “disagree”, “mostly agree “, “agree”, and “strongly agree”, respectively. 

The questionnaire was also used to measure the demographic characteristics of the subjects, including: gender, education level, age, and average income of household members. In order to verify the difference between those who installed household PV and those who did not, we designed a dummy variable: “household PV installed vs. not installed”, denoted by 0 and 1, where 0 means that the resident did not install and 1 means that the resident installed, and named it whether installed (WI). This variable also represents the willingness of residents to install PV in their homes. 

### 3.5. Data Sources

Data collection is conducted offline and online. Team members first selected Jieshang Village, Dongdi Township, Bijie, Guizhou Province, and Gongshang Village, Wuchongan Town, Qian’an County, Hebei Province for offline questionnaire distribution. The reason why these two villages were selected is that the residents who installed household PV in these two villages are relatively dense, and Dongdi Township is even the local pilot project for PV poverty alleviation [62]. Afterwards, we learned that the local residents who installed PV had established WeChat groups to facilitate communication with each other regarding the use of household PV. Therefore, team members used a Chinese online questionnaire platform called Wenjuanxing to create an online survey. The access link to the online questionnaire is: https://www.wjx.cn/vm/tUBjCZF.aspx# accessed on 19 May 2023. Since the team had established contact with some residential PV users during the research process through face-to-face interviews, we requested these homeowners to share the above-mentioned questionnaire through their WeChat groups and requested other residential PV users to fill it out. This method is known as snowball sampling. The snowball sampling method is very suitable for conducting sampling surveys for special groups [63]. During the snowball sampling process, the questionnaire is not only distributed to residents who have already installed household PV, but also to residents who have not installed it. To encourage answering questions, cash rewards are set up. Given the unique nature of this survey, it is necessary for the participants to have certain family and social life experience. Therefore, we limit the age of the respondents to over 25 years old. Due to the use of a snowball method to distribute questionnaires, it is possible for the same respondent to answer two versions of the questionnaire at the same time. Therefore, the research team will screen out these questionnaires by detecting the IP address. If two questionnaires are answered with the same IP address, the two questionnaires will be invalidated. The questionnaire collection work lasted from March to mid-May 2023, and a total of 1430 questionnaires were collected. After eliminating the cases of large omissions, only selecting the same option, and answering two questionnaires with the same IP address, 1249 valid questionnaires were finally retained, with an effective rate of 87.35%. This study essentially requires a comparative analysis of participants who have installed household PV and those who have not. Therefore, we also conducted sample balance tests on the collected samples. The statistical characteristics and balance test results of the samples are shown in Table 1. The balance test was conducted using the *χ*^2^ analysis method, and the test results showed significant differences in demographic characteristics between the uninstalled and installed sample groups. Therefore, it is necessary to incorporate demographic statistical features into the research model and control the variable WI. To prevent valuation bias, we also included demographic characteristics as control variables in the model to control the target variable GC of this study. Based on the synthesis of Section 3.1, Section 3.2 and Section 3.3 in this paper, the conceptual model of this study is shown in Figure 1. 

## 4. Result and Discussion

### 4.1. Reliability and Validity Test of External Model

This study used PLS-SEM to estimate the structural equation model shown in Figure 1. According to the PLS-SEM testing process [64], the external model needs to be tested first, including the reliability and validity test. 

Reliability refers to the relationship between the indicators (scale items) and latent variables (also known as constructs), that is, whether the indicators can be used to reflect latent variables. At this time, four indicators need to be measured: (1) Cronbach’s α; (2) Composite reliability (CR); (3) Average extraction variance (AVE); (4) Factor loadings. Firstly, we use Cronbach’s α to test the various indicators of the external model and measure the reliability of the indicators. Usually, the value of Cronbach’s α is between 0 and 1. If the α coefficient does not exceed 0.6, the internal consistency reliability is generally considered to be insufficient; reaching 0.7–0.8 indicates that the scale has considerable reliability, and reaching 0.8–0.9 indicates that the scale has very good reliability [65]. Secondly, we measure the composite reliability (CR), which is the combination of variable reliability and represents the internal consistency of measures of latent variables. The higher the CR value, the higher the internal consistency of measures of latent variables. Thus, 0.7 is an acceptable threshold [66]; Thirdly, we measure the AVE value, which is the variance explanatory power of the measures of latent variables, also known as convergence validity. The higher the value, the higher the reliability and convergence validity of the measures of latent variables. Ideally, the standard value should be greater than 0.5, and 0.36–0.5 is the acceptable threshold [67]. Finally, it is necessary to measure factor loadings. In the reflective construct, the statistical significance of factor loadings represents the contribution of each question item to a latent variable, which is also between 0 and 1. The higher the value, the better the measurement effect of the measurement indicator on the latent variable. Generally, a value greater than 0.5 is an acceptable range [68]. As shown in Table 2, all indicators in this study meet the above requirements, indicating that the reliability of the external model in this study is acceptable. 

Discriminant validity is the extent to which a construct is truly distinct from other constructs by empirical standards. In the modeling process, a set of scale items cannot reflect both the A construct and the B construct. There are generally three methods for measuring discriminant validity [69]: (1) Cross loadings; (2) Fornell–Rocker criterion; (3) HTMT (heterotrait–monotrait ratio). The measurement principle of method 1 is: calculate the factor loadings for all measures (scale items) on all latent variables, in which the factor loadings of the measurement indicators and their corresponding latent variables are called factor loadings, and the factor loadings generated by the measurement indicators and other latent variables are called cross loadings. When the cross loading is less than the factor loading, it indicates differential validity. The advantage of this method is that its principle is simple and easy to understand, but it has rarely been presented in recent papers. The main reason is that it requires a relatively large table to be presented, occupying the paper layout, and the amount of information is too scattered, requiring readers to gradually compare factor loadings with cross loadings [70]. The discrimination principle of method 2 is: whether the average extraction variation (AVE) of the latent variables is greater than the square of the correlation coefficient between the latent variables and other latent variables, that is, whether the square root of the average extraction variation of the construct is greater than the correlation coefficient between the construct and other constructs. If it is greater, it indicates that it has discrimination validity [67], The results of the Fornell-Rocker criterion test conducted in this study are presented in Table 3 below. Method 3: HTMT (heterotrait–monotrait ratio), which is the ratio of between-trait correlation to within-trait correlation. It is the ratio of the mean value of index correlation between different latent variables to the mean value of index correlation between the same constructs. Henseler et al. believe that if all values in Table 4 are less than 0.85, the structural model can be considered to have discriminative validity [71]. Since Henseler et al. argue that the drawback of the Fornell–Larcker Criterion is the tendency to overestimate the square root of AVE, making it easy to pass discriminant validity tests. Therefore, the commonly used testing methods in practice are the Fornell–Larcker Criterion combined with the HTMT method. The results of both tests are presented in Table 3 and Table 4. Table 3 shows the results of the Fornell–Larcker Criterion, where the diagonal values are the square root of AVE, which are greater than the other values in the table. Table 4 shows the results of the HTMT test. All values in the table are less than 0.85, so both the Fornell–Larcker Criterion and the HTMT results show that the test of discriminant validity can be passed. In conclusion, the test for the relationship between the measures and the constructs in this study passed. 

### 4.2. Structural Test of Inner Model

The structure test of inner model refers to the test of the relationship between latent variables in Figure 1, as well as the test of the hypothesis proposed in 3.1. The final results are presented in Table 5 which is divided into six parts: (1) Single path hypothesis test; (2) Total effect hypothesis test for GE; (3) Total effect hypothesis test for GC; (4) Role of demographic characteristics on WI; (5) Effect of demographic characteristics on GC; (6) Role of marked variables. 

(1). The results of the single path hypothesis test show that H1a, H1b, H1c, H1d, H1e, H1f, and H1g can be supported, indicating that government incentive policies and environmental publicity policies have a certain promoting effect on residents’ choice to install household PV. Users with experience in installing and using household PV have a significant odds ratio in specific user experience (perceived value, perceived risk), green self-efficacy, and green consumption behavior compared to non-installed residents. From the perspective of the stimulating effects of policies on residents’ installation of household PV, the effect of economic incentive policies is significantly greater than that of environmental publicity policies (βH1a > βH1b), indicating that the main driving factor for residents to install household PV is still the stimulation of interests. Residents can experience both perceived value and perceived risk after installing household PV, and the odds ratio of both perceptions is similar, ranging from 0.20 to 0.25 (βH1c = 0.217, βH1d = 0.233), indicating that residents feel both benefits and risks in the process of installing and using PV. From a specific numerical perspective, the odds ratio of perceived risk is slightly greater than that of perceived benefit, indicating that residents still have significant concerns about the future of using household PV. This concern can be attributed to a lack of confidence in product quality, as well as concerns about future policy uncertainty. From a macro perspective, the widespread use of household PV among residents in China began in 2017 [7], and has been ongoing for less than a decade now. It is a relatively new product, and the potential benefits and risks it can generate need to be tested over time. Residents who install household PV can generate green self-efficacy (H1e, H1f are significant) after obtaining richer perceived value and perceived risk, which means that installing and using household PV is a practical activity and a form of learning, that is, the “learning by doing” form. This form can harvest corresponding knowledge, skills, and values, thereby further forming green self-efficacy. Users have improved their evaluation of their ability to execute environmental behavior, strengthening human confidence in managing environmental issues. Specifically, we have found that perceived value has a much greater impact on improving green self-efficacy than perceived risk (βH1e > βH1f). The explanation is that perceived value and perceived risk represent the relationship between economic and environmental benefits. Perceived value represents a pattern where personal and environmental interests can coexist. In this case, after installing household PV, on the one hand, it is beneficial to environmental protection, and on the other hand, it also gains various benefits. This is a win-win model, which shows that the sustainable growth model that takes into account both personal interests and environmental protection interests can gain more recognition from residents, and this model is more conducive to improving the green self-efficacy of residents. Perceived risk represents a model where personal interests are incompatible with environmental interests, meaning that pursuing environmental interests requires a certain amount of personal risk. In this case, it indicates that residents need to bear a certain amount of risk to achieve environmental benefits when installing household PV, which is a mutually exclusive model. Although this model can also improve residents’ sense of green efficacy, it is not as effective as the win-win model. Therefore, seeking a mutually beneficial development model that integrates personal and environmental interests is the sustainable development path that humanity needs to pursue, and it is also more easily accepted by the general public. Finally, green self-efficacy does have a positive impact on improving green consumption (H1g significant), confirming the conclusion of Chen et al. [36]. 

(2). The results of total effect hypothesis test for GE show that H2a, H2b, and H2c can be supported. The support for H2a indicates that residents’ installation and use of household PV can indeed improve their sense of green efficacy, confirming that the process does have a “learning by doing” effect. This indicates that residents’ installation and use of household PV is not only environmentally friendly from the perspective of energy production, but also a learning process. The support for H2b and H2c indicates that the policies currently formulated for the promotion of household PV also have a certain “derivative effect”. Economic incentives are more likely to increase residents’ green self-efficacy than environmental advocacy policies, suggesting that policies that are compatible with both personal and public interests are more likely to be favored by individuals, leading to better derivative effects. 

(3). The results of total effect hypothesis test for GE show that H3a, H3b, and H3c can be supported. The support for H3a indicates that users who install and use household PV do have a higher level of green purchasing compared to those who do not. Combined with the results of H2a hypothesis test, it can be understood that the process of installing and using household PV is a pro-environmental practice, and this practice activity has a long-term nature due to the durable nature of household PV. The long-term practice process can help users generate green self-efficacy and improve their green purchasing level. The support for H3b and H3c indicates that the promotion policies of household PV not only have a derivative effect on improving users’ sense of green efficacy, but also have a “ripple effect” [22] on green purchasing behavior. This “ripple effect” takes the specific experience of residents using household PV and the sense of green efficacy as the dual intermediary. Specifically analyzing the “ripple effect” of the two policies, we found that the ripple effect of economic incentive policies was higher than that of environmental protection publicity policies, which again showed that policy options compatible with individual interests and public interests were better than mutually exclusive options. 

(4). The role of demographic characteristics on WI. From the perspective of regression technology, the main significance of this section is to prevent bias in regression results due to differences in demographic characteristics between the two sets of data in comparative experiments. Moreover, the decision to adopt household PV is a family behavior, not an individual behavior. However, the family is also composed of individuals, so we can only interpret it from the individual’s attitude towards household PV. We found a negative relationship between gender and education level and WI. In this case, 0 represents males and 1 represents females; 0 represents college degree and below, 1 represents bachelor degree (including self-study), and 2 represents graduate degree and above. The regression results indicate that women hold a more negative attitude towards installing household PV than men. It is possible that from a personality perspective, women are relatively more conservative than men, and as an emerging technology product, PV are less accepted by women than men. People with higher levels of education may consider issues more comprehensively and analyze the pros and cons from multiple perspectives such as costs, benefits, and risks, so they are more cautious when dealing with new things. 

(5). From the perspective of the effect of demographic characteristics on green consumer behavior, only income has a significant impact, and the regression results are positive correlation. The explanation is that green consumer behavior needs to pay more economic costs and action costs than ordinary consumers under the current conditions, and residents with higher income are more willing to pay additional costs for environmental protection. 

(6). The marked variables are mainly used to test and control for common method variation (CMV). From the test results, it can be seen that there is no common method variation in this case, which may be due to the cash reward set in this questionnaire survey. Some studies have shown that appropriate rewards can improve the reliability and validity of the questionnaire survey [72]. 

### 4.3. Discussion

This study is similar to the current emerging research on pro-environmental spillover effects, but the pro-environmental spillover studies emphasize the experimental properties of the study, often using different versions of questionnaires for testing. The specific experimental methods include recall methods [73], information feedback methods [74], etc., which emphasize the use of information interventions to prompt participants’ past pro-environmental behaviors and thus further analyze whether the behaviors can influence their future decisions. The methodology used in this paper emphasizes the difference in green self-efficacy and green consumption between household PV users and non-household PV users in the natural state, without designing information intervention items in the questionnaire. The findings from this natural survey approach are closer to the residents’ usual living conditions, and the research findings have more external validity. 

The analysis of policy effects on pro-environmental spillover effects emphasizes the information feedback method, which uses monetary information feedback and environmental information feedback in questionnaires to compare the effects of different information feedback on pro-environmental spillover effects. In conclusion, it is often believed that monetary incentives are more detrimental to the occurrence of pro-environmental spillover effects compared to environmental promotion policies. This analysis tends to divorce the pre-behavior from the spillover behavior. In fact, pro-environmental spillovers can only occur when a policy is effective in promoting the actual occurrence of pre- behavior. In reality, a large number of studies have proven that economic incentives are the most important reason for the occurrence of pre-behavior [7], and without the occurrence of pre-behavior, there can be no post-behavior (spillover behavior). Isolating post-behavior as irrelevant to pre-behavior and exploring how policies that incentivize pre-behavior affect post-behavior can easily lead to metaphysical fallacies. In this case, the analysis of policy effects focuses on the analysis of action paths. The policy variables used in this case are substitution variables, and it is emphasized that the policy variables first affect the pre-behavior (installation and use of household PV), and then have an impact on the post-behavior (green purchasing). The study found that the total effect of economic incentive policies on post-behavior (green purchase) is greater than that of environmental protection publicity, and we found that this effect is based on the premise of having an effect on residents’ installation decisions first. This analysis method regards the pre-behavior and post-behavior as a whole, taking into account the real scene of events, improves the external validity of the study, and can make a useful supplement to the existing literature. 

### 4.4. Limitation and Futurework 

This study also has several limitations. Although this research employs comparative analysis to verify that residents who install household PV have a higher odds ratio of green self-efficacy and green consumption compared to non-installers, it is important to note that other factors such as individual values, social norms, and personality traits [75,76,77] have been shown to influence individuals’ pro-environmental behavior. Unfortunately, due to the constraints of this study, these factors were not included in the analysis. In future studies, further research on these elements will be pursued. Additionally, our sampling method employed online questionnaires and snowball sampling, which may inadvertently exclude individuals such as middle-aged and elderly individuals who are not accustomed to answering surveys online [78]. This limitation undermines the representativeness of our sample. In future research, we aim to employ both online and offline questionnaires, as well as interviews, to gather a comprehensive range of opinions. This approach will ensure enhanced representativeness in our study, allowing for a deeper and more extensive exploration of the topic. 

## 5. Conclusions

This paper employs structural equation modeling to investigate the impact of household PV installation on green consumption. Additionally, it examines the potential “derivative effect” and “ripple effect” of PV promotion policies. The main conclusions are as follows: Firstly, the installation and utilization of residential PV systems can trigger a “learning by doing” effect. This effect assists residents in cultivating green self-efficacy and enhancing their level of green consumption. Secondly, the existing photovoltaic (PV) promotion policies, including economic incentive measures such as subsidies and environmental advocacy policies, have the potential to enhance the adoption of residential PV systems. Moreover, these policies can also cultivate a sense of green self-efficacy among residents, leading to an increased level of green consumption. This ripple effect can further contribute to the overall sustainability goals. Thirdly, economic incentive policies exert a more profound influence on encouraging residents to adopt PV systems, enhancing their belief in their ability to engage in sustainable behaviors, and elevating their level of environmentally friendly consumption, in contrast to environmental protection publicity policies. This indicates that approaches that integrate personal interests with environmental considerations are more likely to yield substantial leverage effects. The above conclusion provides insights into the three questions posed in the introduction. Firstly, residents who have installed and utilized household PV systems exhibit significantly higher odds ratios in green self-efficacy and green purchasing, compared to residents lacking such experience. Secondly, this disparity primarily stems from the “learning by doing” phenomenon. This suggests that residents progressively develop self-efficacy for green practices through practical experience with household PV systems. Consequently, they are more inclined to engage in environmentally conscious purchasing behavior due to the influence of their acquired green self-efficacy. Thirdly, based on the research findings, it is evident that the current PV promotion policy in China effectively encourages residents to adopt green consumer behavior by facilitating the installation of household PV systems. 

This study highlights the additional social value derived from individual environmental protection practices, taking into consideration a comprehensive and perceptive standpoint. This approach surpasses the limited perspective of previous studies that focused solely on individual behaviors. Furthermore, in practical terms, it will assist policymakers in identifying opportunities for policy innovation and broadening the reach of “behavioral leverage”. Our research has several implications for current environmental management. Firstly, we recommend maintaining a stable subsidy policy for household PV. In the promotion process, emphasis should be placed on economic incentives, supplemented by environmental publicity. This policy combination not only promotes the adoption of household PV, but also indirectly fosters other environmentally friendly practices among residents. Additionally, it is crucial to recognize the influential role of household PV in shaping users’ pro-environmental behavior. Environmental promotion should be reoriented towards environmental training, which incorporates user education on the utilization of household PV systems, placing emphasis on enhancing users’ green self-efficacy. Specific environmental knowledge and skills have the potential to enhance individuals’ green self-efficacy, rather than merely promoting the importance of environmental conservation. Moreover, our findings suggest that policies and practices that consider both personal interests and environmental concerns are more likely to positively influence individuals’ levels of green self-efficacy, consequently shaping their behavior towards environmental protection. Therefore, we contend that the future government policies should focus on cost reduction for residents’ pro-environmental activities. One possible approach is to introduce environmental protection labels, which will aid residents in identifying green products. Additionally, implementing a system to trace and identify the carbon footprint of goods can further contribute to minimizing the cost associated with residents’ identification efforts. 

## Figures and Tables

**Figure 1 behavsci-13-00612-f001:**
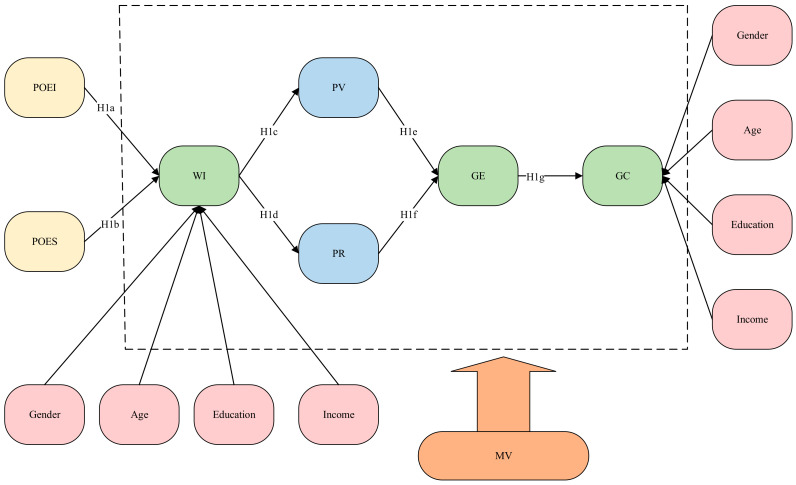
Conceptual Model. The dashed line represents the endogenous variables of this study, all of which are controlled by MV.

**Table 1 behavsci-13-00612-t001:** Sample statistical characteristics and balance test results.

Category	Options	Uninstalled Sample (n = 538)		Installed Sample (n = 711)		*χ*^2^/F–Test
		Number of People	Proportion	Number of People	Proportion	
Gender	Male	225	41.80%	431	60.60%	43.40 (***)
	Female	313	58.20%	280	39.40%	
Age	25–30 years old	238	44.20%	200	28.10%	78.25 (***)
	31–36 years old	146	27.10%	363	51.10%	
	37–42 years old	107	19.90%	119	16.70%	
	Over 43 years old	47	8.70%	29	4.10%	
Educational Background	College degree and below	344	63.90%	520	73.10%	13.25 (***)
	Bachelor degree (including self-study)	159	29.60%	164	23.10%	
	Graduate degree and above	35	6.50%	27	3.80%	
Average Annual Income of Household Members	Under 12,000	118	21.90%	132	18.57%	17.76 (***)
	12,001–30,000	142	26.40%	186	26.16%	
	30,001–50,000	212	39.40%	249	35.02%	
	50,001–100,000	38	7.10%	101	14.21%	
	Above 100,001	28	5.20%	43	6.05%	

*** *p* < 0.001.

**Table 2 behavsci-13-00612-t002:** Reliability and convergence testing of external models.

Latent Variables	Items	Mean (SD)	Factor Loading	Cronbach’s α	Composite Reliability	AVE
POEI	POEI1	3.38 (1.374)	0.803	0.794	0.872	0.695
	POEI2	3.275 (1.378)	0.903			
	POEI3	3.348 (1.407)	0.791			
POES	POES1	3.011 (1.53)	0.895	0.841	0.903	0.757
	POES2	3.062 (1.415)	0.852			
	POES3	3.199 (1.383)	0.862			
PV	PV1	2.879 (1.355)	0.800	0.828	0.886	0.659
	PV2	2.840 (1.442)	0.818			
	PV3	2.926 (1.374)	0.820			
	PV4	3.122 (1.318)	0.810			
PR	PR1	2.866 (1.383)	0.855	0.814	0.876	0.64
	PR2	2.907 (1.381)	0.831			
	PR3	2.966 (1.407)	0.817			
	PR4	3.00 (1.152)	0.687			
GE	GE1	3.165 (1.341)	0.817	0.885	0.92	0.743
	GE2	2.935 (1.526)	0.902			
	GE3	3.038 (1.367)	0.865			
	GE4	2.998 (1.301)	0.862			
GC	GC1	2.886 (1.445)	0.817	0.841	0.894	0.677
	GC2	2.996 (1.452)	0.814			
	GC3	2.785 (1.535)	0.833			
	GC4	2.792 (1.522)	0.829			
MV	MV1	3.16 (1.329)	0.835	0.815	0.89	0.73
	MV2	3.255 (1.359)	0.862			
	MV3	3.189 (1.414)	0.866			

**Table 3 behavsci-13-00612-t003:** Fornell–Larcker criterion.

Variables	POEI	POES	PV	PR	GE	GC	MV
POEI	0.832						
POES	0.063	0.870					
PV	0.111	−0.020	0.812				
PR	0.127	−0.110	−0.054	0.800			
GE	0.171	−0.060	0.440	0.178	0.862		
GC	0.112	0.079	0.098	−0.022	0.201	0.822	
MV	−0.021	−0.018	−0.008	0.006	0.023	−0.034	0.844

The values on the diagonal in Table 3 are all square roots of the AVE values. In this table, due to the fact that the values in the upper-right corner are the same as those in the lower-left corner, only the values in the lower-left corner are shown.

**Table 4 behavsci-13-00612-t004:** Heterotrait–monotrait ratio.

Variables	POEI	POES	PV	PR	GE	GC	MV
POEI							
POES	0.074						
PV	0.118	0.060					
PR	0.148	0.135	0.074				
GE	0.200	0.075	0.510	0.201			
GC	0.128	0.123	0.129	0.078	0.226		
MV	0.036	0.043	0.023	0.014	0.028	0.042	

All values in the table below 0.85 indicate that the model in this paper has good discriminant validity.

**Table 5 behavsci-13-00612-t005:** Hypothesis test results.

Hypothesis	Path	Standardized Path Coefficient	*t*-Value	Support or Not
Single path hypothesis test				
H1a	POEI→WI	0.171	6.409 (***)	Y
H1b	POES→WI	0.061	2.236 (*)	Y
H1c	WI→PV	0.217	7.742 (***)	Y
H1d	WI→PR	0.233	8.815 (***)	Y
H1e	PV→GE	0.449	21,286 (***)	Y
H1f	PR→GE	0.203	8.214 (***)	Y
H1g	GE→GC	0.203	7.436 (***)	Y
Total effect hypothesis test for GE				
H2a	WI (T)→GE	0.145	9.064 (***)	Y
H2b	POEI (T)→GE	0.024	4.926 (***)	Y
H2c	POES (T)→GE	0.010	2.302 (*)	Y
Total effect hypothesis test for GC				
H3a	WI (T)→GC	0.030	5.792 (***)	Y
H3b	POEI (T)→GC	0.005	4.278 (***)	Y
H3c	POES (T)→GC	0.002	2.223 (*)	Y
Role of demographic characteristics on WI				
Gender→WI		−0.161	5.611 (***)	Y
Age→WI		0.036	1.229 (NS)	N
Education→WI		−0.096	3.504 (***)	Y
Income→WI		0.063	2.225 (*)	Y
Effect of demographic characteristics on GC				
Gender→GC		0.056	1.755 (NS)	N
Age→GC		0.005	0.179 (NS)	N
Education→GC		−0.006	0.207 (NS)	N
Income→GC		0.08	2.65 (**)	Y
Role of marked variables				
MV→PV		−0.013	0.367 (NS)	N
MV→PR		0.005	0.141 (NS)	N
MV→GE		0.025	0.945 (NS)	N
MV→GC		−0.043	1.383 (NS)	N

* *p* < 0.05, ** *p* < 0.01, *** *p* < 0.001; NS, no significant effect.

## Data Availability

The data in this study involves third-party privacy and is supported by the Youth Social Science and Humanities Fund of the Ministry of Education of China. The data is shared with third parties and is not convenient for disclosure.

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
