# Peer review of "Does the Pro-Environmental Behavior of Household PV Installation Contribute to the Shaping of Users’ Green Purchasing Behavior?—Evidence from China"

_behavsci, 2023, doi:10.3390/bs13070612_

Round 1

Reviewer 1 Report

The aticle is very interesting and well stuctured. In addition to the "learn by doing" model I suggest to consider "material participation" as proposed by Marres (2012) to support the argumentation.

Marres, N. (2012), Material Participation: Technology, the Environment and Everyday Publics, palgrave macmillan

Minor remarks:

what does it mean: "Per NEA," (line 45)

change "govt." to governental (line 51)

add references to quoted terms as "ripple effect" (line 455)

Reviewer 2 Report

This paper mainly explores the incentive effect of house hold PV installation on green consumption, and discusses the possible "derivative effect" and "ripple effect" of the PV promotion policies. On the whole, the paper is interesting and well-organized. However, there are still some issues to be solved, which have been listed as follows.

1. The introduction needs to be reworked. The importance of promoting house hold PV installation, the importance of green consumption, and the specific links between the two need to be considered.

2. The literature review is small and incomplete. It is necessary to add the relevant literature on this topic.

3. In 3.2 Formulation of research hypothesis, the author only gets conclusions from the existing literature and lack the thought process to derive them.

4. The author should check "4. Literature Review", this title whether has writing errors?

Minor editing of English language required.

Reviewer 3 Report

This is an interesting study discussing how the pro-environmental behavior of household PV installation contribute to the shaping of users' green purchasing behavior

 The paper is well-written but few issues require authors' attention.

please refer to the comments below to further improve the work and make it ready for publishing.

Abstract

Please add statistical evidence of your findings to support your results with numbers

P 2 line 51..govt.?

CO2..subscipt

Literature review section requires further attention. Currently it is very summarized not covering essential background information required for this study

Please revise the research hypothesis as they seem very similar in p 4 line 176, 177

‘H1a: The POEI can positively promote the installation of household PV by residents

H1b: The POES can positively promote the installation of PV’

P 4 line 181.. [33,34]and…leave a space

[11,35] In this study.. missing full stop

Please revise the following hypothesis as it does not make good sense compared to the one before ‘H1f: Perceived risk in user experience can enhance users' sense of green efficiency’. 

The sources in table 1 can be enriched to support the model structure

Please add more details about the survey data, channel…etc.

Please make a grammar check, for example ‘ the 280 research team will screen out these questionnaires by detecting the IP address’, why use the future tense for a work which was already completed. please perform proof reading and language check.

Please separate the discussion section from the conclusion section. The former should include the research limitations and discuss the results obtained compared to previous ones. The latter should relate the aims and objectives of the study and highlight the research novelty.

It is overall well-written but with few grammatical errors which requires proof reading and language check.

Reviewer 4 Report

The manuscript concerns the important issue of the investigation of the development of green consumerism via household PV installation. To accomplish the "dual carbon goal," the Chinese government is encouraging the adoption of household photovoltaic (PV) systems. While previous studies have examined people's willingness to install PV, there has been limited focus on how PV installation and usage influence environmentally friendly behaviors. Thus, this study aims to explore the impact of household PV installation on the development of green consumerism. By applying the "learning by doing" principle, a survey was conducted with 1249 participants, and a generalized structural equation model was used for analysis. The research findings indicate that installing and using household PV systems contribute to green consumption, and the current PV promotion policies indirectly support environmentally friendly consumption. Moreover, economic incentive policies have a greater influence than environmental publicity policies. Based on these results, several recommendations are proposed: 1. Maintaining consistent economic incentives to promote household PV adoption; 2. Prioritizing the dissemination of knowledge and skills for promoting environmental protection; 3. Striving to align personal interests and societal interests with low-carbon policies. Remarks and comments: The section on Analytic Hierarchy Process (AHP) basics should be moved to the section of the appendix. Line 163: The choice of reference should be supplemented with respect to the government subsidies have a positive impact on the adoption of household PV by residents, eg. Most Searched Topics in the Scientific Literature on Failures in Photovoltaic Installations. Energies 2022, 15, 8108. https://doi.org/10.3390/en15218108. Why this particular approach was chosen for the analysis? This should be underlined in the general framework of the presented study. Why this approach is the best solution for performed analysis? Are there concrete steps that can be recommended and how generalizable are the findings? Can they be applied to other areas? How dependent are they to specific characteristics of the region under examination? The conclusion should be interpreted and deliver the meanings, this section should be more focused and based on the results. Are there concrete steps that can be recommended? What is the novelty of the distinguished method. 

Round 2

Reviewer 2 Report

I'm glad my advice can help you. It can be seen that compared with the previous version, this version has made a lot of adjustments. I think these adjustments have responded well to the suggestions. 

The language of this paper needs to be further polished. Avoid Chinglish.

Reviewer 3 Report

Thank you for your detailed responses to the raised comments. The responses are satisfactory but please make one final language check and proof reading because I have noticed some mistakes e.g. missing full stops at the end of sentences, and missing spaces.

 I have noticed some mistakes e.g. missing full stops at the end of sentences, and missing spaces.

Reviewer 4 Report

To accomplish the "dual carbon goal," the Chinese government is encouraging the adoption of household PV usage. The study aims to explore whether the environmentally conscious behavior resulting from household PV installation contributes to the development of users' green purchasing habits. Remarks: The abbreviation of the latent variable perception of economic incentives occurs before its explanation, please add it earlier. Please check the other variables.
